# Enhancing Large Language Models for Constraint-Driven Molecular Generation and Beyond

## Abstract

Most de novo molecule generators attempt to satisfy hard chemical constraints in a single forward pass, offering little guidance when outputs fall short. We introduce Code-Driven Molecular Synthesis (CDMS) – an iterative, model-agnostic framework that embeds a formal self-improving feedback loop into large language models (LLMs). At the start of each task, the LLM uses the chemist's request as input to generate a snippet of executable code, referred to as an *inspector*, which formalizes the evaluation logic for guiding molecular refinement. This inspector remains fixed throughout the refinement process and is executed on every candidate molecule at each iteration. It produces natural-language critiques describing how to improve the molecule to better meet user-defined constraints (e.g., "add a para-hydroxyl group"). These *Programmatic Feedback Gradients* are appended to subsequent prompts, guiding the LLM toward progressively refined outputs until all structural and functional requirements are satisfied. CDMS achieves state-of-the-art success rates in constraint satisfaction using only a few feedback iterations and without any model retraining. To encourage further research, we release a benchmark dataset curated for code-generated, feedback-driven molecular design [1].

## 1 Introduction

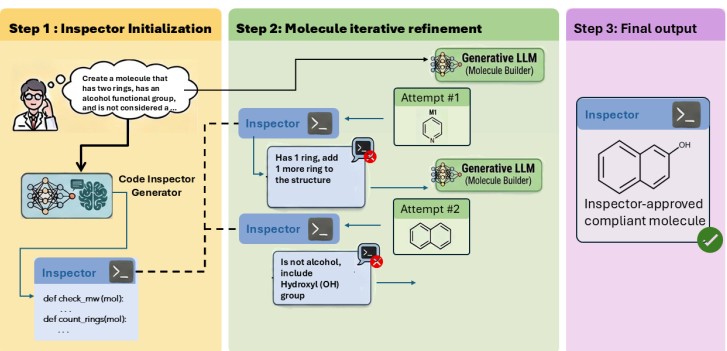

Figure 1: CDMS molecule generation process

Chemical synthesis is a complex and multi-stage process. It begins by identifying the desired attributes of the chemical compound to be synthesized, including its molecular structure, properties, and potential applications. Initially, chemists create base molecules, which are then extensively tested for properties such as synthetic feasibility and application-related attributes, like drug-likeness, toxicity, etc. Recent advancements have introduced end-to-end generative models to streamline molecular candidate generation. However, these

---

[1]https://anonymous.4open.science/r/CDMS-C08D/

models often generate molecules that do not adhere to the chemists' requirements or are impractical to synthesize due to unavailable precursors or overly complex synthesis routes. Therefore, in practice, the synthesis pipeline is split into two separate stages: generating molecules with high potential for meeting the specified constraints, followed by filtering these candidates using classifiers and other models, to ensure high synthesizability, and optimize other domain-specific attributes like toxicity, drug-likeness, etc. This paper focuses on addressing the first stage of generating molecular candidates that adhere to the desired constraints. We demonstrate that CDMS achieves high synthesizability scores and introduce expert-tailored datasets specifically designed for this task.

Recently, models that leverage LLMs have shown state-of-the-art (SOTA) performance in fields like natural sciences and chem-informatics, including applications in *de novo* molecule generation Li et al. (2023); Zhang et al. (2024); Arraf & Radinsky (2024). While LLMs demonstrate a broad understanding of chemical concepts in the molecule synthesis domain, their ability to meet multiple and complex, user-defined constraints remains limited largely due to their inability to apply their knowledge. Generating molecules that satisfy functional, structural, and dynamic requirements—critical for drug discovery or materials science—poses a significant challenge for these models. For instance, generating a molecule with multiple constraints, such as two aromatic rings, a molecular weight of $(150, 300)$ Da, and a carboxylic acid group, requires precise control over functional group placement, counting of ring systems, and physical properties—an area where current models often fall short.

We address these challenges with a novel framework composed of two components: a generative model that designs molecules and an inspector that validates compliance with user-defined textual constraints via executable code. The inspector plays a central role in an iterative feedback loop by generating *Programmatic Feedback Gradients* – natural-language signals derived from code execution that guide the model toward satisfying the specified requirements. When constraint violations are detected, such as incorrect stereochemistry or invalid ring structures, the inspector returns actionable textual feedback, which is incorporated into subsequent prompts, progressively refining the generated molecules toward full compliance. Figure 1 illustrates the end-to-end feedback loop, and an example of the molecular revision process is shown in Figure 2. This mechanism ensures robust alignment between generated outputs and the target properties.

We collaborated with chemistry and computer science experts to construct a unique dataset of molecular generation constraints, along with corresponding golden inspector code capable of verifying whether generated molecules satisfy the specified requirements. Empirically, we demonstrate that CDMS outperforms state-of-the-art (SOTA) models for molecule synthesis, including both traditional approaches and recent LLM-based generation and refinement methods. Moreover, we provide evidence that CDMS is broadly applicable beyond molecular design, showing its effectiveness in general natural language processing (NLP) tasks that require strict, constraint-driven text generation.

The contributions of this work are threefold: **(1)** We introduce a novel framework that combines generative models with a novel Inspector Model capable of producing executable code to iteratively validate outputs against complex requirements. We also show that CDMS can generate molecules that are achieves higher validity rates than previous SOTA models. This iterative feedback loop ensures high precision and robust alignment between generated outputs and user-defined constraints. **(2)** Leveraging an expert-curated dataset, we empirically demonstrate that CDMS outperforms traditional LLMs and SOTA refinement techniques in molecular generation tasks. **(3)** We establish the versatility of CDMS by showcasing its successful application to broader NLP tasks requiring constraint-driven generation, highlighting its potential to generalize beyond molecular design into other domains.

## 2 Related work

**Text-to-Molecule Generation.** Molecule synthesis from textual descriptions is a key area in molecular generation. Early work used SMILES (Simplified Molecular Input Line Entry System) representations — a textual format for encoding molecular structure- with BioT5+ Pei et al. (2024a) training a transformer on the ChEBI-20 dataset Edwards et al. (2021) that links molecular descriptions with structures.

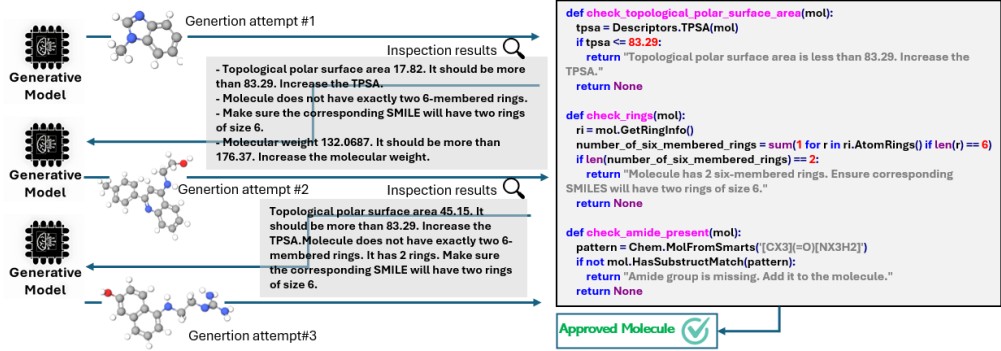

Figure 2: Overview of the CDMS iterative refinement approach for molecule generation. The left panel outlines the step-by-step process of refining molecular structures based on textual feedback, while the right panel demonstrates the Inspector model's generated code for validating molecular constraints. This code-driven feedback mechanism ensures alignment with user-defined requirements, enabling precise and iterative enhancements to the generated molecules.

Recent models have advanced alignment and generation strategies. MolReFlect Li et al. (2024) uses in-context alignment, and LDMol Chang & Ye (2024) adopts a diffusion-based latent approach—though both lack public code. MoleculeSTM Liu et al. (2023) enables text-guided molecule editing via contrastive learning in a joint latent space. More recently, mCLM Edwards et al. (2025) proposes a modular chemical language model that represents molecules as synthesis-compatible functional building blocks aligned with natural language, enabling multimodal reasoning over molecular structure and function, but it fails to understand a complex set of constraints for molecule design as we show in Section 5.1.

LLM-based methods such as ChemLLM Zhang et al. (2024) and MolReGPT Li et al. (2023) extend generation through domain specialization and retrieval-augmented generation, respectively. ChatDrug Liu et al. (2024) and ChemCrow Bran et al. (2024) introduce iterative loops using external tools like RDKit to guide improvements, but lack formal validation or enforcement of constraints.

In contrast, **CDMS** dynamically generates a single task-specific *inspector* as executable code, which performs exhaustive, deterministic validation and returns *Programmatic Feedback Gradients.* This enables an iterative refinement loop that continues until all constraints are satisfied—offering stronger compliance guarantees than retrieval- or tool-augmented methods.

**Iterative Refinement.** Iterative refinement is widely used in text generation, initially relying on human feedback Bai et al. (2022); Fang et al. (2023a) and later on scalar rewards Liu et al. (2022) or free-form critiques Madaan et al. (2024); Shinn et al. (2023). However, these methods struggle with complex, multi-constraint objectives.

Molecule optimization has followed similar trends, combining heuristics or expert-driven retrosynthesis Fang et al. (2023a;b), but these approaches lack scalability.

TextGrad Yuksekgonul et al. (2025) incorporates external tools to produce auxiliary validation signals and generates a new verification program at each iteration to evaluate the previous output. This design introduces additional call overhead and may lead to inconsistent inspection across iterations. The resulting signals are then interpreted by an LLM to determine the refinement strategy, while the enforcement of the inferred corrections is delegated to another LLM.

In contrast, *CDMS* compiles task constraints into a persistent executable inspector (executable code) that deterministically verifies constraints and produces explicit corrective prompts as an output of the code directly, eliminating the need for LLM-based interpretation of validation signals at each iteration.

## 3 CDMS Framework

In this section, we present CDMS—an iterative self-refinement methodology designed for the first phase of molecule synthesis, where candidate molecules are generated based on textual requirements. CDMS integrates code-enhanced feedback into the generation process, ensuring alignment with the specified constraints and improving the quality of the generated candidates. CDMS is composed of three distinct models: **(1)** the Memory component, which stores previous outputs and feedback; **(2)** the Generative Model ($M_G$), responsible for primary content generation; and **(3)** and the Inspector Model ($M_I$), which evaluates the generated content using generated code and provides approval or constructive feedback. We would like to emphasize that our model is versatile and can be applied to any generative model as $M_G$. In each iteration, $M_G$ receives the input from both the user (i.e. the set of requirements) and the textual feedback provided by $M_I$. Consequently, $M_I$ takes the content generated by $M_G$ to produce feedback for the next iteration of $M_G$. We show that as long as the LLM understands the aspect to be optimized (e.g., reducing molecular weight translates to reducing molecule count or using lighter molecules) and there is a self-evaluation loop instructing which aspects to optimize, this can lead to more aligned molecules. This section details each component and how they function collaboratively within the CDMS framework. We empirically show that CDMS outperforms previous SOTA models on ChemGen, a dataset we introduce, in addition to two synthetic tasks in natural language (NL) generation aimed to highlight LLM weak points that CDMS can help with.

### 3.1 Generative Model

The generative model can be any language model that supports a turn-by-turn input format. CDMS allows any model to be plugged in, where we show a performance boost in every model $M_G$. In our study, we experiment with foundation models with world knowledge and those designed for chemistry GPT-4.1 OpenAI (2025), GPT-4o OpenAI (2024), LLaMA3.3-70B et al (2024), and Claude3.5 Haiku Anthropic (2023), while we also leverage more specialized models in the framework of CDMS, such as MolReGPT Li et al. (2023).

### 3.2 Inspector Model

The *Inspector* model ($M_I$) is implemented as an NL-to-code generation system, specifically designed to generate code that evaluates whether a given SMILES representation of a molecule adheres to a set of constraints provided by the user in NL. The model is configured to decompose complex molecular requirements into smaller sub-requirements, enabling a fine-grained systematic evaluation of the molecule's compliance.

The inspector generates the inspection code only once during the inference process, when obtaining user requirements. Unlike previous methods, this code can then be optionally reviewed by a human-in-the-loop and saved to $M_I$'s state and is reused at every iteration. At each iteration, $M_I$ inspects the SMILES formula against these requirements by executing the inspector code with the SMILES as a string input, identifying any violations, and producing *programmatic actionable feedback* as seen in Figure 2 for refinement. This iterative feedback mechanism is intended to ensure that the generated molecule progressively aligns with the specified constraints. Our implementation uses GPT-4o as the primary code generation backbone due to resource availability, with additional experiments utilizing alternative models detailed in Section 5.5.2.

#### 3.2.1 Code Generation and Execution

The model receives a prompt to generate Python code, i.e., the *Inspector code*, that checks whether a given set of constraints is satisfied. If a constraint is not met, the generated code function should return a list of textual feedback describing the violation and the necessary modifications for each violation.

The generated code follows a format where each constraint is implemented as a single method, ensuring modularity and clarity. A single entry point orchestrator function is responsible for running these inspector methods and returning aggregated results, maintaining consistency, and reducing variability. For reproducibility and further exploration, we provide the prompt used in Appendix A.

### 3.2.2 Feedback Generation

The code generation model is instructed to generate an entry point method that encapsulates other method calls. CDMS invokes the entry point method with the generated molecule SMILES and receives the list of feedback, then compiles it back to a textual format to be passed to $M_G$.

### 3.2.3 Enhancing Code Reliability

To minimize the variability of the generated code and improve precision, the code generation model is supplied once with a single targeted few-shot example. This example is an illustration of what we expect the code to do and how to structure it.

We draw the reader's attention to the fact that this example is supplied to the code generation model as a few-shot and is not tailored per task. This example is important in guiding the code generation process, ensuring that the generated code structure reflects intended functions and conforms to required formats with less reliance on extensive prompt engineering. Given these measures and considering the clear, easy-to-read textual format of the input, we did not observe significant instances of inaccurate code that justify further action.

### 3.3 Memory

The memory component is essential for preserving the context of interactions between the generative model and the inspector model. It serves as a dynamic log, recording each attempt by the generative model and the corresponding feedback from the inspector. This log links all prior interactions, allowing the generative model to reference and learn from past failures, similar to revisiting an advisor's feedback. The generative model uses this memory as input, we use the memory as in-context learning to prevent the generative model from repeating previous mistakes. The data in this memory is organized in JSON format as seen in Figure 3

```
[
{"role":  "user", "content":  "<task specification>"},
{"role":  "assistant", "content":  "<candidate output>"},
{"role":  "user", "content":  "<inspector feedback>"},
{"role":  "assistant", "content":  "<revised candidate output>"}
...
]
```

Figure 3: Structure of the memory buffer that records generator outputs and inspector feedback during iterative refinement.

### 3.4 Iterative Algorithm

The generation process is outlined in Algorithm 1. The inspector and generator arrangement establishes an adversarial yet constructive feedback loop. In this loop, the interaction is akin to a dialogue from the perspective of the generative model, which retains the memory of previous exchanges. This memory aids the generative model in avoiding past mistakes and preventing the repetition of errors. The code generation model, on the other hand, is specifically tuned to the numerical and structural attributes of the text. It generates feedback that the generative model uses to refine its outputs. This dynamic ensures that the generated content is not only accurate but also contextually relevant.

Please note that upon exceeding the maximum number of optimization attempts, CDMS outputs the last molecule generated, together with a signal if the inspector approves it.

## 4 Empirical Evaluation

In this section, we present the empirical evaluation framework, along with the baselines, datasets, and methods used to assess our model.

---

**Algorithm 1** CDMS algorithm

---

1: **Input:** NLP molecule requirements.
2: **Output:** Content adheres to the constraint.
3:
4: inspector_code ← Inspector(constraints)
5: feedback ← constraint
6: attempt count $n \leftarrow 0$
7: **while** $n <$ maximum optimization attempts **do**
8:    response ← GenModel(feedback)
9:    feedback ← inspector_code.exec(response)
10:    **if** feedback is approval **then**
11:       **return** approved content
12:    **end if**
13:    $n \leftarrow n + 1$
14: **end while**
15: **return** response

---

## 4.1 Datasets

Datasets in de novo molecule generation such as Text2Mol Edwards et al. (2021) and LlaSMol Yu et al. (2024) typically define a single "golden" target molecule because the natural language description specifies a single exact molecular structure. In contrast, our setting describes a *set of constraints* that a valid molecule must satisfy. For a given set of attributes, many molecules may satisfy the requirements. Consequently, datasets designed for single-target reconstruction are not suitable for evaluating constraint-driven molecule generation. To address this limitation we introduce **ChemGen**, a dataset for generating molecules from natural language requirements expressed as sets of molecular constraints.

ChemGen adopts a one-to-many evaluation paradigm: multiple molecules may satisfy the same requirement set. Instead of comparing generated molecules to a single ground-truth structure, evaluation verifies whether the generated molecule satisfies all constraints. To enable this evaluation, each requirement is paired with an expert-authored executable *validation program* ("golden code") that deterministically checks constraint satisfaction.

## 4.2 Dataset Construction

The dataset was constructed through collaboration between chemists and computer scientists. Chemists curated a diverse collection of molecular properties commonly studied in chemical compound databases such as PubChem Kim et al. (2023). These include physicochemical attributes (e.g., molecular weight and topological polar surface area), structural properties (e.g., ring counts and rotatable bonds), and functional group requirements describing chemical reactivity.

Computer scientists implemented executable validators for each requirement using code-based routines that compute molecular descriptors and verify constraint satisfaction. For every requirement set, chemists identified at least one molecule from PubChem satisfying the constraints, ensuring that each entry corresponds to a chemically realizable compound.

Natural language requirements follow a structured constraint language that combines molecular properties, comparison operators, and functional group patterns.

## 4.3 Dataset Statistics

ChemGen contains 779 unique samples and a total of 4,599 constraints. Each compound is associated with a set of 3–9 requirements (5.9 on average), reflecting the constraint-driven formulation of the task.

The constraints span multiple molecular property categories and comparison operators. In total, the dataset includes molecular properties, such as ring counts, covalent bond counts, molecular weight, topological polar surface area (TPSA), heavy atom counts, rotatable bonds, and functional group requirements. These properties are expressed through ten comparison operator types (e.g., inequalities, bounded ranges, equality relations, and presence or absence constraints). Across the dataset, the natural language descriptions are generated from 53 unique requirement templates (after abstracting numerical values), and functional group detection is implemented using 18 SMARTS patterns corresponding to chemically meaningful substructures. This structure allows ChemGen to represent a diverse range of molecular requirements while maintaining consistent evaluation through executable validation code.

Finally, we analyze how restrictive the constraint sets are by measuring how many molecules in PubChem satisfy the specified requirements. PubChem currently contains approximately 115 million compounds, providing a large reference space for evaluating constraint difficulty. For many entries in ChemGen, the requirements narrow this space to only a small number of candidates. In 0.4% of the dataset, only one molecule in the entire PubChem database is expected to satisfy the constraints, indicating extremely restrictive requirement sets. Only 8.4% of entries correspond to more than 1,000 candidate molecules in PubChem, despite the database containing over 115 million compounds. Overall, these statistics demonstrate that ChemGen does not consist of trivial constraint descriptions: many requirement sets isolate only a very small fraction of the molecules in a database of more than 115 million compounds, making the generation task significantly constrained.

## 4.4 Baselines

We evaluate CDMS against several baselines, including zero-shot generative models, iterative refinement models, and task-specific trained models. We use GPT-4.1, GPT-4o, LLaMA-3.3-70b, and Claude 3.5 for the underlying generative models. Additionally, we experiment with advanced prompt engineering techniques, such as chain-of-thought reasoning and few-shot generation, to enhance performance by structuring tasks more effectively and providing examples for context.

For iterative refinement, we select Self-Refine Madaan et al. (2024), which employs iterative refinement with textual self-feedback generated by the LLM to progressively improve model outputs. We additionally evaluated TextGrad Yuksekgonul et al. (2025), which incorporates external tools such as RDKit Landrum & contributors (2024) in an iterative, tool-guided generation setup. However, the original TextGrad work does not report empirical results on molecule generation performance at all so we benchmark it on ChemGen. For a fair comparison, we adapted the CDMS feedback prompt to TextGrad's input format. Both TextGrad and Self-Refine were allowed the same number of refinement iterations during evaluation.

We benchmark specialized models: ChemLLM Zhang et al. (2024) as an LLM dedicated to chemistry and trained on molecule generation, and mCLM Edwards et al. (2025), a modular chemical language model that jointly models natural language and molecular structures by representing molecules as synthesis-aware functional building blocks rather than atom-level tokens.

We finetuned BioT5+ Pei et al. (2024b), a model trained on the ChEBI-20 Shardlow et al. (2018) and considered SOTA on Text2Mol dataset Edwards et al. (2021) for text-to-molecule tasks. Appendix B provides more details about the fine-tuning process. we draw the reader's attention to the fact that ChemGen does not have an aligned golden-standard molecule that intuitively supports fine-tuning, but used the ChemGen raw data (see Section 4.1).

We also include MolReGPT Li et al. (2023), the SOTA generative model designed to create molecular structures based on textual descriptions. In the analysis below, we refer to it as MolRe-*X* for generality.

To demonstrate the versatility of CDMS as a framework compatible with various models, we incorporate CDMS using MolReGPT as the underlying generative model.

We use GPT-4.1, GPT-4o, LLaMA-3.3-70B, and Claude 3.5 Haiku as the main generator backbones in our experiments. Later we refer to CDMS as the configuration with CDMS frameworks leveraging MolReGPT with GPT-4o as its generative backbone.

Some specialized molecule generation models (e.g., mCLM, ChemLLM, BioT5+) are evaluated only in a standalone setting. These models do not support multi-turn or feedback-conditioned generation, and therefore cannot be integrated into the CDMS refinement loop.

### 4.5 Metrics and Evaluation

We evaluate the performance of CDMS and baseline models (see Section 4.4) on our custom dataset (see Section 4.1). The evaluation metric is the percentage of outputs generated by each model that comply with the specified molecular requirements.

## 5 Empirical Results

This section presents our findings and ablation studies, analyzing the contributions of the coder and text generation models to CDMS performance.

### 5.1 Main Result

| Category | Framework | GPT-4.1 | GPT-4o | LLaMA-3 | Claude 3.5 |
|---|---|---|---|---|---|
| **LLM Prompting Strategies** | Zero-Shot | 36.30 | 25.42 | 22.72 | 7.06 |
| | Chain-of-Thought (CoT) | 31.10 | 29.41 | 37.25 | 12.58 |
| | Few-shot | 32.60 | 28.23 | 36.17 | 13.09 |
| | CoT + Few-shot | 37.08 | 30.84 | 39.01 | 16.17 |
| **Molecular Design Strategy** | MolRe-*X* Li et al. (2023) | 38.20 | 43.13 | 35.49 | 23.62 |
| | ChemLLM Zhang et al. (2024) | ———— 5.59 (standalone) ———— | | | |
| | mCLM Zhang et al. (2024) | ———— 3.9 (standalone) ———— | | | |
| **Iterative Feedback** | Self-Refine Madaan et al. (2024) | 60.0 | 26.83 | 24.39 | 15.01 |
| | TextGrad Yuksekgonul et al. (2025) | 61.3 | 25.83 | 28.17 | 21.13 |
| | k-attempts ($k = 7$) | 37.0 | 26.24 | 23.82 | 7.19 |
| **Finetuned model** | BioT5+ | ———— 25.93 (standalone) ———— | | | |
| **CDMS** | Zero-Shot | 63.10 | 35.30 | 34.19 | 23.74 |
| | CoT* | 64.81 | 39.15 | 52.94 | 24.00 |
| | Few-shot | 72.67 | 41.91 | 48.00 | 24.18 |
| | CoT + Few-shot | 83.70 | 48.09 | 53.92 | 25.80 |
| | MolRe-*X* | **86.35** | **51.47** | **54.34** | **35.04** |

Table 1: Performance of models with different optimization approaches. Greyed rows indicate standalone results that do not rely on LLMs.

Table 1 presents a comparative analysis of task-expert models, baseline LLMs and numerous prompting and refinement approaches. In this experiment, for a given set of textual constraints, the baselines were tasked with generating a molecule that adheres to chemical requirements. Results in bold show the winner in each model column, and all results in bold are statistically significant[2].

We observe that GPT-4.1 is the strongest GPT-based model and achieves particularly high performance under iterative feedback strategies. LLaMA-3 generally outperforms GPT-4o and Claude 3.5 in standard prompting settings, especially in methods like Chain-of-Thought (CoT) and CoT + Few-shot. MolRe-X, a molecular design model, provides a strong specialized baseline, as expected from a task-specific model.

The CDMS strategy consistently delivers the best results, regardless of the baseline model used (GPT-4.1, GPT-4o, LLaMA-3, or Claude 3.5) or the prompting strategy. Notably, MolRe-X also achieves its highest

---

[2]Statistical significance was assessed for all baselines using McNemar's test on paired binary outcomes ($\alpha = 0.05$); results were deemed significant when $p < 0.05$

performance under the CDMS strategy, demonstrating its effectiveness in molecular design tasks. This trend holds for all models tested, where the iterative textual feedback improves upon any optimization strategy.

We observed that an LLM with k-attempts, where multiple outputs are generated simultaneously without interconnection, yields lower results compared to CDMS. This highlights the importance of the feedback mechanism, as opposed to relying solely on multiple iterations. We also observe the superiority of our model compared to other textual feedback models that are not based on code, such as Self-Refine. Additionally, we experimented with fine-tuned BioT5+, which achieved a non-trivial standalone result. However, its reliance on a large amount of annotated and aligned data raises concerns about its scalability.

We note that although TextGrad generates code at each iteration and is computationally more exhaustive, CDMS outperforms it with a single comprehensive inspector with the executable textual gradient.

## 5.2 Number of Refinement Attempts

We analyzed the intermediate outputs of CDMS with GPT-4o to determine the percentage of results, whether accepted or rejected by the inspector, that are validated by the golden validator. The results, presented in Figure 4, highlight the performance of the two generator backbones analyzed in this refinement-depth study, GPT-4o and LLaMA-3. We demonstrate that the quality of results improves as the number of refinement iterations increases, with optimal performance at 7 attempts. This analysis over a separate validation set was also instrumental in tuning the hyperparameter for the number of refinement iterations. We draw the reader's attention that these are not the performance results of CDMS but the intermediate results that are yet to be accepted or rejected by the inspector model.

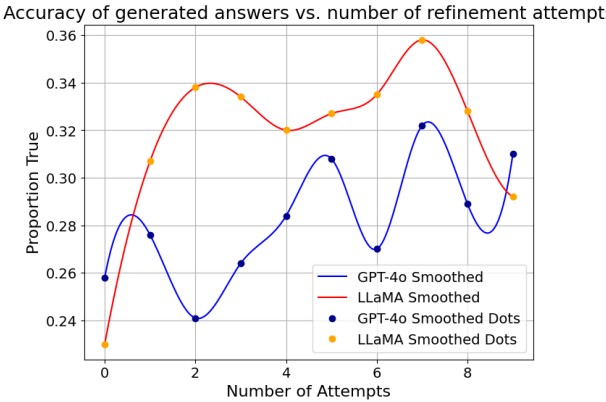

Figure 4: CDMS quality results in improvement vs. refinement attempts *not* including the last iteration result approved by the inspector. Note that these results are lower than CDMS's performance results, as they include generation attempts unapproved by the inspector as well.

## 5.3 Molecular Novelty and Diversity

We compared our generated results with the PubChem molecule dataset Kim et al. (2023) and found that 73.77% of the generated molecules were not present in the dataset, demonstrating that CDMS acts as a molecule generator rather than merely retrieving known compounds, highlighting its potential in drug discovery. In addition, we analyzed the structural variation among the generated molecules and found an average molecular diversity (Tanimoto-based) of 0.898, compared to 0.903 for the original molecules from which the ChemGen dataset was constructed. This indicates that CDMS maintains a high level of chemical diversity while satisfying complex constraints.

## 5.4 Inspector Quality

In this ablation, we vary only the inspector-generation model, while keeping the generator fixed (GPT-4o). To evaluate the impact of the inspector component's coder model in CDMS, we perform an ablation study

using generative models known to produce lower-quality code. As shown in Figure 5, the overall accuracy of CDMS degrades with weaker models. Notably, even when using GPT-4o mini—which demonstrates poor performance on the SWE-benchmark Jimenez et al. (2024)—CDMS maintains competitive performance. In fact, CDMS with GPT-4o mini slightly underperforms MolReGPT, the next best model, despite relying on a substantially weaker code generator.

### 5.5 Ablation Tests

To evaluate the effectiveness of each component in our framework, we conducted the following ablation tests. For ablations that isolate a single component, we fix the generator to GPT-4o to maintain comparability with the original experimental setup and vary only the component under study.

#### 5.5.1 Binary vs. Text Feedback

We aim to test the type of code used to generate feedback. Specifically, we will evaluate the impact of codes that generate textual feedback versus those that produce simpler binary inspectors, which return a binary signal based on whether the inspector approves or rejects the proposed molecule. We ran the experiment with two representative high-performing backbones from the original ablation setting. The results presented in Table 2 clearly show the effectiveness of the textual feedback in CDMS.

| Methodology | Binary Feedback | | CDMS | |
|---|---|---|---|---|
| | GPT-4o | LLaMA-3 | GPT-4o | LLaMA-3 |
| CoT | 32.22 | 30.03 | **39.15** | **52.94** |
| Few-shot | 30.16 | 37.09 | **41.91** | **48.00** |
| CoT + Few-shot | 31.06 | 40.82 | **48.09** | **53.92** |

Table 2: Performance of CDMS with binary feedback.

#### 5.5.2 Inspector Code Generation Model

We evaluated the impact of the code generation model (see Section 3.2) on overall CDMS performance by replacing the inspector-generation model while keeping the rest of the pipeline fixed. Figure 5 compares the resulting CDMS accuracy against each model's SWE-bench Jimenez et al. (2024) performance. Overall, stronger code models are presumed to yield better inspectors and we observe higher downstream CDMS accuracy. Although the relationship is not strictly monotonic, we note a trend of lower CDMS performance when coupled with weaker performing models on SWE-bench. In particular, LLaMA 3.3 achieved the best CDMS accuracy ($\sim$55%), outperforming GPT-4.1 despite lower SWE-bench performance. Most inspector-generating models also surpassed the MolRe-X baseline, while all of them substantially outperformed BioT5+. These results indicate that the quality of the generated inspector code is an important factor in CDMS performance, but general-purpose coding ability alone does not fully determine downstream effectiveness.

## 6 Applications of CDMS

### 6.1 Application to Drug-like Generation

We compare (i) a one-shot generator that directly maximises QED and (ii) a two-step, lab-style pipeline: heuristic candidate generation (e.g., limited ring count, no reactive groups) followed by QED filtering. The former explores a wider chemical space yet often proposes unsynthesizable molecules; the latter narrows the search but boosts synthetic accessibility.

To quantify the trade-off, we compare state-of-the-art end-to-end molecule generators against a two-stage variant whose first step is either CDMS or one of the molecule-generation baselines considered in this setting. After enforcing expert chemical constraints (Appendix C), we rank candidates by QED and keep the top

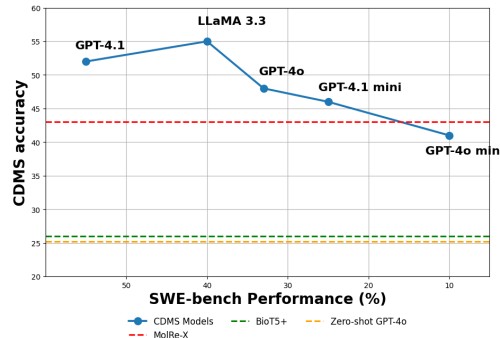

Figure 5: CDMS performance with weaker coder models.

| Category | Generative method | SAS (↓) | QED (↑) | XLogP |
|---|---|---|---|---|
| **End-to-End** | BioT5+ | 6.14 | 0.325 | 3.92 |
| | MolReGPT | 2.94 | 0.493 | 4.21 |
| **Two-Stage** | BioT5+ | 3.57 | 0.554 | **2.71** |
| | MolReGPT | 2.38 | 0.756 | 3.81 |
| | CDMS | **2.12** | **0.790** | 3.83 |

Table 3: Comparison of end-to-end and two-stage molecule generators across SAS, QED, and XLogP. CDMS was not explicitly prompted to optimize for XLogP, and therefore does not exhibit bias toward this property. Its unbiased performance on XLogP emerges from only satisfying broader user constraints. In contrast, other models were trained on datasets biased toward soluble molecules, which may explain their consistently lower XLogP scores.

10 %. The molecule sampling and post-generation selection procedure used in this experiment is detailed in Appendix K. Table 3 shows that CDMS delivers both the lowest SAS (best feasibility) and the highest QED, outperforming BioT5+ and MolReGPT in either setting. We attribute this to the iterative process that builds the molecule gradually. We also note that XlogP (a measure of water solubility) shows that CDMS yields unbiased molecules, as it was not asked to generate soluble molecules, whereas other models produced significantly more soluble compounds — a bias that is not always desirable depending on downstream tasks. Additional optimization results are in Appendix D.

## 7 Generalizing to Other NLP Tasks

Our method extends beyond molecular generation to tasks like constrained text generation Lin et al. (2020). To demonstrate this, we designed two new synthetic datasets aimed at addressing LLM limitations in numerical reasoning, structural planning, and constraint adherence. **TextGen** focuses on textual content generation with multi-level constraints, such as ensuring paragraph, sentence limits, and word properties. For example, a task might require creating *"a document with three paragraphs, the first and last being equal in length, and sentences with fewer than 10 words"*. These tasks test the model's compliance with intricate requirements in text. **StructuredGen** targets structured data generation, requiring models to produce outputs like nested objects or datasets based on detailed specifications. For instance, a task might involve *"generating a list of dictionaries with specific key-value formats and relationships"*. For more examples and details on the generation and structure of these datasets, please see Appendix E. Same as in ChemGen, these datasets include a golden inspector code that validates whether a given string adheres to the corresponding textual constraints.

These datasets challenge LLMs to handle both textual and data-oriented reasoning in a controlled, constraint-driven context.

### 7.1 Results

We benchmarked CDMS with various underlying generative models against other iterative refinement methods on TextGen and StructuredGen and without iterative refinement. Our results in Table 8 show a significant improvement over baseline models in both these tasks as well. By employing CDMS, we outperform Self-Refine—a text-to-text feedback generation method using LLMs for intermediate reasoning—our approach and Reflexion Shinn et al. (2023), which uses reward feedback. Results show a consistent quality improvement trend, with 15%-44% gains over the baseline. The quality improvement trend observed in TextGen and StructuredGen was observed across more baselines, please see Appendix F.

## 8 Conclusions

In this work, we present a novel framework that significantly advances the field of molecular generation by integrating LLMs with an innovative feedback mechanism using Programmatic Feedback Gradients. This iterative feedback loop, powered by an Inspector Model that generates executable validation code, ensures precise compliance with complex, user-defined molecular constraints. Our results demonstrate that CDMS outperforms traditional LLMs and SOTA refinement methods in tasks such as de-novo molecular synthesis, where adherence to multifaceted structural, functional, and dynamic requirements is critical. We contribute our code and dataset to the community for further research.

For future work, we aim to explore the broader applicability and generalizability of the CDMS framework across various domains. Additionally, we wish to expand CDMS to seamlessly integrate new functionalities, such as instructing the LLM to perform tasks as they are detected. For example, in a molecule reaction-chain detection framework, the LLM could be taught to search for specific chain lengths and provide actionable feedback, all without the need for fine-tuning or additional data.

## 9 Limitations

Our method does not directly validate whether the generated code fully captures the intended requirements. Nonetheless, even imperfect code significantly outperforms existing models. Empirically, we found that the generated code enhances LLM performance by providing valuable, targeted feedback. Manual validation revealed that 96% of inspector code snippets accurately represent the requirements expressed in the original natural language prompts. Further discussion on improving code stability and alignment is provided in Section 3.2.3.

One limitation of CDMS is its linear time complexity with respect to the number of refinement attempts (see Section 3). We believe this can be mitigated through more rigorous prompting strategies, potentially reducing the number of required iterations (see Section 3.2.3). However, as drug discovery is typically not time-sensitive, we consider runtime a secondary concern and prioritize the generation of constraint-compliant molecules. This also marks a shift from traditional iterative refinement, as the feedback in each iteration is not generated by a language model but is instead the result of code execution, enabling more deterministic and verifiable corrections. A second limitation is that CDMS does not guarantee molecular stability. While the generated molecules may meet specified constraints, they are not necessarily chemically stable. This can be addressed by incorporating domain-specific checks—such as identifying weak bonds or unstable substructures—into the inspector logic. We leave this for future work. A third limitation is that CDMS cannot validate properties that lack a computable definition. While this constrains the range of enforceable requirements, we conducted an expert review to assess the framework's practical coverage. Specifically, we asked a domain expert to evaluate which synthesis-relevant properties—drawn from PubChem Kim et al. (2023), a public NIH database of chemical structures and properties, and from prior works on molecule synthesis Fu et al. (2021); Pei et al. (2024b)—can be encoded in executable code. The expert determined that approximately 50% of key attributes are directly computable, and an additional 25% can be estimated using code-accessible deep learning models. The remaining 25% lack reliable computational methods and were excluded to avoid introducing low-confidence signals. This analysis suggests that CDMS can enforce nearly 75% of synthesis-relevant properties without requiring lab-based validation, highlighting its practical applicability in computational molecule design. A full breakdown is provided in Appendix L.

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

## A   Prompt and Output Template

We offer a look inside the hood of CDMS. In this work, we leverage foundation models as a code generation model for the inspector component and as a generative model. For reproducibility and transparency, we share the prompts used in our framework. The prompts can be seen per model in Table 4.

We also share the specific examples used to tune the inspector model.

```python
from rdkit import Chem
from rdkit.Chem import Descriptors

def check_molecular_weight(mol):
    mw = Descriptors.rdMolDescriptors.CalcExactMolWt(mol)
    if mw < 100:
        needed_increase = 100 - mw
        return f"Molecular weight is {mw:.2f}, which is below 100. Increase it by
            adding heavier functional groups or atoms to increase by at least {
            needed_increase:.2f} Da."
    if mw > 200:
        needed_decrease = mw - 200
        return f"Molecular weight is {mw:.2f}, which exceeds 200. Decrease it by
            removing or replacing functional groups or atoms to decrease by at
            least {needed_decrease:.2f} Da."
    return None

def check_number_of_atoms(mol):
    num_atoms = mol.GetNumAtoms()
    if num_atoms < 10:
        needed_atoms = 10 - num_atoms
        return f"Number of atoms is {num_atoms}, which is less than 10. Add at
            least {needed_atoms} more atoms."
    if num_atoms > 50:
        excess_atoms = num_atoms - 50
        return f"Number of atoms is {num_atoms}, which exceeds 50. Remove at least
            {excess_atoms} atoms."
    return None

def validate_smiles(model_output):
```

| Component | Content |
|---|---|
| **Generative model** | *You are a skilled chemist tasked with the creation of a new drug.* |
| | *I will provide you with a list of specific constraints, and your objective is to generate a molecule that meets all of these criteria.* |
| | *There is always a feasible solution.* |
| | *You must deliver the result as a SMILES string, ensuring that the molecule you propose is a real, chemically valid compound.* |
| | *MOST IMPORTANTLY: You must provide only the SMILES string as the output, without any additional characters or information.* |
| **Inspector** | *You are a code generation model for a chemist researching components. Your task is to generate code from a natural language constraint on molecules that allows me to validate if a certain text complies with the constraints in the prompt.* |
| | *The code should have one entry point - compile_feedback that takes in a string and calls all other constraint-checking functions. This function should return a list of the feedback messages from all the constraint-checking functions, and clean the list from None values.* |
| | *Each constraint-checking function should return a string with textual feedback if the constraint is not met. The textual feedback should describe how the answer violates the constraints this function is checking and how the model can check it. If the text complies, the function should return None.* |
| | *Make the code return descriptive, actionable textual feedback for each case of the violation and describe how exactly the model should change the output to comply with the constraints.* |
| | *Include this feedback method always to validate the SMILES of the component.* |
| | *Make sure you don't add any example inputs or instance answers in the code.* |
| | *If the SMILES is invalid, return only the message of invalid SMILES format as feedback. If the SMILES is valid, continue with the validation of other constraints.* |
| | *Write only the code as a string - do not include any comments or explanations in the code. I will use the code for my purposes.* |

Table 4: Inspector model and generative model prompts.

```
25      mol = Chem.MolFromSmiles(model_output)
26      if mol is None:
27          raise ValueError("Invalid SMILES format.")
28      return mol
29
30  def check_alcohol_present(mol):
31      pattern = Chem.MolFromSmarts('[OX2H]')
32      if not mol.HasSubstructMatch(pattern):
33          return "Alcohol group (OH) is missing. Consider adding an OH group to the
                molecule."
34
35  def compile_feedback(model_output):
36      try:
37          mol = validate_smiles(model_output)
38          if mol is None:
39              return ["Invalid SMILES format. Generate it again to proceed with the
                    validation of other constraints."]
40      except Exception as e:
41          return [f"Parsing error of the SMILES here is why: {str(e)}. \nGenerate it
                again to proceed with the validation of other constraints."]
42
```

```
43      constraints = [check_number_of_atoms , check_molecular_weight ,
            check_alcohol_present]
44      feedback = [constraint(mol) for constraint in constraints]
45      # Remove None values
46      inspection = [message for message in feedback if message is not None]
47      return inspection
```

## B  Baseline Training

As mentioned in section 4.1, ChemGen has input requirements in natural language, while the target is to generate a molecule that complies with the golden-inspector in each entry. Having this format, ChemGen does not naively support training a model to generate molecules from the textual description. To fine-tune BioT5+, we relied on the raw collected data that ChemGen was created from – where scraped data about a given molecule has been collected and processed to resemble the input format in ChemGen, where the target was the SMILES of the molecule. Using this aligned data, we trained BioT5+. We trained the proposed BioT5+ for 5 iterations on 60% of our dataset, with 20% of it for evaluation, and the rest used for testing.

## C  ChemGen Dataset

Our dataset comprises a set of constraints that a chemist might consider when designing a molecule to satisfy specific propertiesGuo & Schwaller (2024). The dataset was constructed using real-world, well-researched molecules through the following steps:

1. **Defining Desired Attributes:** We collaborated with chemists to identify a set of desired attributes that are commonly of interest in molecular design.

2. **Collecting Molecules:** A total of 779 randomly selected molecules were collected from the PubChem database, covering multiple domains such as salts, industrial building blocks, drug discovery, acids, and more.

3. **Attribute Annotation:** For each molecule, we retrieved the studied attributes (see Appendix C.1) annotated by human domain experts in the PubChem database. From this, we filtered attributes that matched our predefined set of studied properties.

4. **Generating Text Descriptions:** Human annotators compiled textual descriptions of the attributes observed in each molecule. For example, a molecule with 2 rings of sizes 2 and 3 and a carboxyl functional group might be described as: *"This molecule has 2 rings, one of size 2 and the other is of size 3, and has a carboxyl functional group."*

5. **Code-Based Verification:** Computer scientists annotated these textual constraints to create a *golden-inspector* – a code-based validator. This tool verifies whether a given molecule, represented as a SMILES string, adheres to the textual constraints defined by the dataset.

6. **Further Validation:** To further validate the annotated code, we used the source molecule from which the text description—and consequently the code—was generated. We ran the golden-inspector code on this source molecule to ensure there were no errors in dataset creation and to verify that at least one molecule satisfies the *golden-inspector*. Additionally, we conducted a set of experiments to ensure that the code correctly fails for at least one other molecule from the dataset, thereby minimizing the number of false positives.

Our process ensures the relevancy of the generated constraints and their interest in the domain of de novo molecule generation.

### C.1 Studied Constraints

This section outlines the key constraints in the dataset and their motivations. The source molecule is annotated by $S$

1. **Molecular Weight (MW):** Defines the range of molecular weights to ensure physical and chemical property consistency, critical for solubility, stability, or activity.

$$\Big[0.8MW(S), 1.2MW(S)\Big]$$

2. **Heavy Atoms (HA):** Limits the number of non-hydrogen atoms to control molecular complexity, relevant for drug design and material science. The limit was set as

$$\Big[max\big(0, HA(S) - 2\big), HA(S) + 2\Big]$$

3. **Topological Polar Surface Area (TPSA):** Constrained to predict membrane permeability and solubility, ensuring suitable pharmacokinetic profiles. We set the TSPA as

$$\Big[0.8TPSA(S), 2TPSA(S)\Big]$$

4. **Covalent Bond Count (CB):** Specifies the total number of covalent bonds between atoms in the molecular graph. This constraint helps control molecular size, structural complexity, and connectivity.

$$\Big[0.8CB(S), 1.2CB(S)\Big]$$

5. **Rotatable Bonds(RB):** Controls molecular flexibility for balanced rigidity and adaptability, essential for ligand-receptor interactions.

$$\Big[0.8RB(S), 1.5RB(S)\Big]$$

6. **Rings and Ring Molecules:** Constrains cyclic structures, ensuring structural diversity and alignment with desired properties.

$$\text{rings number} = \Big[max(0, Rn(S) - 1), Rn(S) + 2\Big]$$

$$\text{ring size} = \Big[max(0, Rs(S) - 2), Rs(S) + 1\Big]$$

7. **Functional Groups:** Enforces inclusion/exclusion of specific functional groups to tailor compounds for various applications. Supported groups include:

   - **Alcohol:** -OH (hydroxyl group)
   - **Carboxylic Acid:** -COOH (carboxyl group)
   - **Ketone:** >C=O (carbonyl group)
   - **Aldehyde:** -CHO (formyl group)
   - **Amine:** $-NH_2$/-NH- (primary/secondary amine)
   - **Amide:** $-CONH_2$ (amide group)
   - **Ester:** -COO- (ester group)
   - **Ether:** -O- (ether group)
   - **Thiol:** -SH (thiol group)
   - **Alkene:** C=C (double bond)
   - **Alkyne:** C≡C (triple bond)

- **Phenol:** -$C_6H_4OH$ (phenolic group)
- **Halide:** -X (halogen atom, where X = F, Cl, Br, or I)
- **Nitrile:** -CN (nitrile group)
- **Sulfonic Acid:** -$SO_3H$ (sulfonic acid group)
- **Sulfoxide:** >SO (sulfoxide group)
- **Phosphate:** -$PO_4$ (phosphate group)
- **Nitro:** -$NO_2$ (nitro group)

These constraints ensure dataset relevance for targeted applications while maintaining diversity and utility. We contribute our source code and dataset to the community for further research (here).

## C.2 ChemGen Examples

Below, we showcase several samples from our dataset. These examples demonstrate how natural language molecule requirements can be translated into corresponding Python code snippets. Each snippet illustrates the implementation of a specific requirement. Please see Table 5 and Table 6 for two examples.

# D CDMS for Drug Like End-to-End Molecular Generation

## D.1 CDMS to optimize drug-like aspects

We conducted an experiment to demonstrate that CDMS, with $M_G$ =GPT-4o, can optimize specific attributes of generated molecules when prompted. To do this $M_I$ was supplied with external prediction tools to incorporate as inspectors. We incorporated into the inspector model's prompt how to use the tool, while the evaluation was delegated to the specialized model.

In this experiment we asked the model in the form of text to optimize attributes in the molecule such as synthesizability (SAS), drug likeness (QED), and solubility in water (LogP). The target was to optimize beyond the average achieved without filtering by CDMS. The full results are shown in Table 7. BioT5+ and MolReGPT were not designed to optimize particular molecular aspects, and as a result, we observed no notable improvements when textual optimization was attempted via text requirement which led to their exclusion from this table. CDMS, however, successfully optimized each aspect when using the predictor as a code inspector within the optimization process. Regarding constraint adherence accuracy, we did not observe any degradation. We note that the additional constraints led to an increase in the allowed number of optimization attempts, which was set to 10 to compensate for the more complex constraints.

| Generative Model | SAS(↓) | | QED(↑) | | LogP(↓) | |
|---|---|---|---|---|---|---|
| | Avg | Top 10% | Avg | Top 10% | Avg | Top 10% |
| CDMS (SAS opt) | 3.09 | **1.39** | 0.44 | 0.71 | 1.39 | -0.39 |
| CDMS (QED opt) | **2.91** | 1.83 | **0.51** | **0.75** | 4.98 | 0.92 |
| CDMS (LogP opt) | 4.10 | 1.79 | 0.419 | 0.57 | **1.09** | **-0.96** |

Table 7: Performance of CDMS under different optimization criteria (SAS, QED, LogP).

## D.2 Generative model SAS quality

We evaluated the SAS scores of the molecules generated by each generative model in Section 6.1. Notably, CDMS demonstrated the ability to consistently generate molecules within the low-SAS region. We notice that BioT5+ mainly produces high-SAS molecules, reflecting the strengths of CDMS.

| Requirement | Code Snippet |
|---|---|
| **1. Covalent bonds count must be less than 9** | ```python
from rdkit import Chem

molecule = Chem.MolFromSmiles(smiles)
result = molecule.GetNumBonds() < 9
``` |
| **2. Topological polar surface area must be greater than 41.206245401121755** | ```python
from rdkit import Chem
from rdkit.Chem import Descriptors

molecule = Chem.MolFromSmiles(smiles)
result = Descriptors.TPSA(molecule) >
    41.206245401121755
``` |
| **3. Molecule must not contain Carboxylic Acid** | ```python
from rdkit import Chem

molecule = Chem.MolFromSmiles(smiles)
pattern = Chem.MolFromSmarts('C(=O)O')
result = not molecule.HasSubstructMatch(
    pattern)
``` |
| **4. Molecule must contain Sulfonic Acid** | ```python
from rdkit import Chem

molecule = Chem.MolFromSmiles(smiles)
pattern = Chem.MolFromSmarts('S(=O)(=O)O'
    )
result = molecule.HasSubstructMatch(
    pattern)
``` |
| **5. Heavy atom count must be greater than 3 and less than 12** | ```python
from rdkit import Chem

molecule = Chem.MolFromSmiles(smiles)
result = 3 < molecule.GetNumHeavyAtoms()
    < 12
``` |
| **6. Rotatable bond count must be less than 1** | ```python
from rdkit import Chem
from rdkit.Chem import Descriptors

molecule = Chem.MolFromSmiles(smiles)
result = Descriptors.NumRotatableBonds(
    molecule) < 1
``` |

Table 5: Requirements and corresponding Python code snippets for molecular validation.

| Requirement | Code Snippet |
|---|---|
| **1. Heavy atom count must be less than 39** | ```python
from rdkit import Chem

molecule = Chem.MolFromSmiles(smiles)
result = molecule.GetNumHeavyAtoms() < 39
``` |
| **2. Molecular weight must be less than 383.4636744835153** | ```python
from rdkit import Chem
from rdkit.Chem import Descriptors

molecule = Chem.MolFromSmiles(smiles)
weight = Descriptors.rdMolDescriptors.
    CalcExactMolWt(molecule)
result = weight < 383.4636744835153
``` |
| **3. Topological polar surface area must be less than 73.27746879710806** | ```python
from rdkit import Chem
from rdkit.Chem import Descriptors

molecule = Chem.MolFromSmiles(smiles)
result = Descriptors.TPSA(molecule) <
    73.27746879710806
``` |
| **4. Molecule must have 1 ring of size 6** | ```python
from rdkit import Chem
from collections import Counter

molecule = Chem.MolFromSmiles(smiles)
rings = molecule.GetRingInfo().AtomRings
    ()
ring_sizes = [len(ring) for ring in rings
    ]
ring_counter = Counter(ring_sizes)
result = ring_counter.get(6, 0) == 1
``` |
| **5. Molecule must not contain Phosphate** | ```python
from rdkit import Chem

molecule = Chem.MolFromSmiles(smiles)
pattern = Chem.MolFromSmarts('P(=O)(O)(O)
    [#6]')
result = not molecule.HasSubstructMatch(
    pattern)
``` |

Table 6: Additional requirements and corresponding Python code snippets for molecular validation.

# E  NLP domain datasets

## E.1  TextGen

The TextGen dataset is designed to evaluate the performance of large language models (LLMs) in numerical reasoning and compliance with textual constraints. For instance, a sample task could be: *"Create a document with **three** sections. The **first** and **last** sections should be of **equal length**, while the **second** section must be at least **twice as long**. Ensure that each **sentence** contains **fewer than 10 words**, and each word has **fewer than 7 letters**."* Such prompts can be employed for generating simplified reading material for children, the use cases are endless. Similar to the ChemGen dataset, TextGen incorporates both hierarchical textual constraints and golden inspectors to verify compliance with these constraints. The constraints are organized across multiple levels:

- **Document level:** Limits overall document length and the number of paragraphs.

- **Paragraph level:** paragraph length and the number of sentences restrictions.

- **Sentence level:** Constraints on the number of words per sentence.

- **Word and sub-word level:** Requirements on word length, part-of-speech, and syllable counts.

To further increase task complexity and assess reasoning over hierarchical constraints, each entity that imposes requirements on its sub-entities employs an aggregation method. The supported methods are as follows:

- **None:** None of the sub-requirements must be met.

- **All:** All sub-requirements must be met.

- **At-least-X:** At least $X$ sub-requirements must be satisfied.These methods challenge the model's ability to reason about and comply with intricate constraints.

The iterative feedback mechanism, initially proposed for CDMS, has also proven to be effective for TextGen. A straightforward prompt modification (see Appendix H for the specific prompts) enables the model to surpass prior SOTA results in content generation while maintaining compliance with the specified constraints.

## E.2  StructuredGen

While the previous dataset (TextGen) focuses on generating text with hierarchical constraints, StructuredGen targets the generation of structured content such as objects, data classes, or entire datasets. For example, a task might require the LLM to generate a structured data type with the following specifications: " *A **list** containing a **dictionary** with: (1) **three** entries where the **key is a float** and the value is a **list of three strings**, and (2) **four entries** where the **key is a string** and the **value is a dictionary** with specific entries.*"

To evaluate LLMs' ability to manage data complexities, we define two primary classes of data types:

1. *Terminal types:* One of [`int`, `float`, `str`].

2. *Complex types:* One of [`list`, `dict`].

As in TextGen, constraints must be satisfied across the data hierarchy, including aggregation requirements on sub-entities. This ensures the generated output adheres to the specified conditions and integrates multiple constraints seamlessly.

| Dataset | Method | GPT4o (1.7T) | LLama-3.1 (405B) | Phi3-medium (14B) | Mistral (7B) | GPT4o with CoT | GPT4o with K-shot |
|---|---|---|---|---|---|---|---|
| Structured-Gen | Base | 31.12 | 34.52 | 15.55 | 12.44 | 31.76 | 33.48 |
| | Self-RefineMadaan et al. (2024) | 32.14 | 35.07 | 20.53 | 13.14 | 33.13 | 33.47 |
| | ReflexionShinn et al. (2023) | 33.51 | 34.73 | 18.13 | 14.41 | 34.47 | 36.36 |
| | CDMS (Ours) | **35.92** | **49.44** | **24.02** | **18.91** | **37.02** | **41.61** |
| TextGen | Base | 53.91 | 49.23 | 28.05 | 30.61 | 56.13 | 55.75 |
| | Self-RefineMadaan et al. (2024) | 54.61 | 50.21 | 40.12 | 31.62 | 57.26 | 58.62 |
| | ReflexionShinn et al. (2023) | 55.23 | 51.24 | 37.34 | 29.55 | 57.14 | 59.52 |
| | CDMS (Ours) | **62.65** | **57.35** | **55.23** | **44.10** | **64.05** | **70.52** |

Table 8: Averaged Performance of Models Across Datasets

## F    TextGen and StructuredData Results

We conducted a human evaluation on a sample of the generated content, asking participants to assess the quality of each passage without being informed of the number of iterations involved in the optimization process. The results indicate that the iterative refinement process does not impact the perceived quality of the generated content. Figure 8 presents the normalized scores assigned by the human annotators to the evaluated models. The human evaluation to assess the quality of the generated text observed no significant deterioration in quality after five iterations—the maximum number of allowed attempts for this task. The human evaluators were graduate and undergraduate students of engineering who volunteered to annotate this task for scientific purposes. The full evaluation prompt was as follows: "Rate the following content based on its logical flow, vocabulary usage, and overall quality. Provide a score between 1 and 5, with 5 being the highest.".

The overall quality measured by the proportion of outputs that would be accepted by the golden inspector is shown in Figure 8.

## G    Natural Language Constrained Generation Dataset

We propose two new datasets to highlight the shortcomings of current LLMs, particularly in numerical reasoning, structural planning, and adherence to complex requirements. These datasets are:

- **TextGen Dataset:** Designed for free-text generation under hierarchical constraints.

- **Structured Content Dataset:** Focuses on generating structured data, such as lists and dictionaries, adhering to specific requirements.

The tasks are split into easy and hard versions to accommodate varying complexity levels. Easy tasks mirror typical day-to-day constraints (e.g., Create a text about flowers with at least 100 words and fewer than 120 words). Hard tasks stress-test models with intricate requirements (e.g., Write a Haiku where each verse starts and ends with the same two letters, with 5 verses mentioning 'tree' exactly once in each).

### G.1    Dataset Versions and Goals

Our datasets serve to:

1. Exemplify challenges LLMs face when handling complex instructions.

2. Showcase how our method overcomes these challenges by generating code to meet stringent criteria.

### G.2 TextGen Dataset

The TextGen dataset evaluates the structural comprehension and numerical reasoning capabilities of LLMs in free-text generation tasks. Examples include generating paragraphs, sentences, and words under specific constraints. For instance:

> Generate a text that satisfies the following constraints:
>
> 1. A paragraph that both starts and ends with the same word.
>
> 2. A paragraph that contains at least three sentences.

Figure 6: Example Text Generation Constraints

### G.3 Structured Content Dataset

The Structured Content Dataset focuses on generating structured data types, such as lists and dictionaries, with specific constraints. For example:

> Generate a structured data type that includes:
>
> - A list containing a dictionary with the following entries:
>   - Three entries where the key is a float and the value is a list containing three strings.
>   - Four entries where the key is a string and the value is a dictionary with specific entries.

Figure 7: Example Structured Data Generation Description

The dataset includes tasks to test LLMs' ability to manage terminal data types (e.g., `int`, `float`, `str`) and complex data types (e.g., `list`, `dict`).

## H  TextGen Prompt

We aimed to evaluate the applicability of CDMS in constrained natural language generation. To this end, we used the same architecture while modifying only the prompting, as detailed in Table 9. Additionally, we included a few-shot example to both the generative model and the inspector model from the dataset to better tune the model.

```
1       [ {"role": "user", "content": <'content'>},
2        {"role": "assistant", "content": <'content'>} ]
```

## I  Model hyperparameter

The hyperparameter tuning process for our model involved extensive experimentation to ensure optimal performance while minimizing reliance on collected data. Below, we summarize the key findings and decisions made during this process:

**Number of Iterations for CDMS**  Initially, the maximum allowed number of iterations for CDMS was set to 20. However, using a validation set (separate from the test set), we determined that the optimal number of iterations is 7.

| Component | Content |
|---|---|
| **Generative model** | *You are a skilled [ **Text generation/Json Data generation** ] tasked with the creation of a new drug [**Data**].*
*I will provide you with a list of specific constraints, and your objective is to generate a [**Data**] that meets all of these criteria.*
*There is always a feasible solution.*
*~~You must deliver the result as a SMILES string, ensuring that the molecule you propose is a real, chemically valid compound.~~*
*MOST IMPORTANTLY: You must provide only the ~~SMILES~~ string as the output, without any additional characters or information.* |
| **Inspector** | *You are a code generation model for a [**Natural Text generation / Json Generation**]. Your task is to generate code from a natural language constraint on molecules that allows me to validate if a certain text complies with the constraints in the prompt. The code should have one entry point - compile_feedback that takes in a string and calls all other constraint-checking functions. This function should return a list of the feedback messages from all the constraint-checking functions, and clean the list from None values.*
*Each constraint-checking function should return a string with textual feedback if the constraint is not met. The textual feedback should describe how the answer violates the constraints this function is checking, and how the model can check it. If the text complies, the function should return None.*
*Make the code return descriptive, actionable textual feedback for each case of the violation and describe how exactly the model should change the output to comply with the constraints.*
*Include this feedback method always to validate the SMILES of the component.*
*Make sure you don't add any example inputs or instance answers in the code.*
*If the SMILES is invalid, return only the message of invalid SMILES format as feedback. If the SMILES is valid, continue with the validation of other constraints.*
*Write only the code as a string - do not include any comments or explanations in the code. I will use the code for my purposes.* |

Table 9: Inspector model and generative model prompts in TextGen and structured data generation tasks.

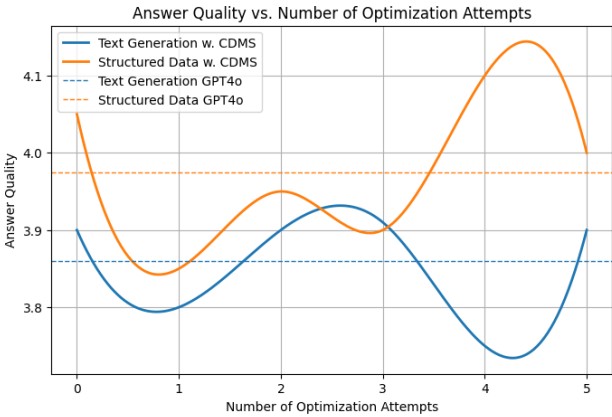

Figure 8: Normalized content quality of generated answers after a number of optimization attempts

**Few-Shot Examples**  We experimented with the number of few-shot examples provided to the model. The goal was to identify the minimal number of examples required to achieve the best results while reducing dependency on collected data. For the inspector code generation model, where the model generates code from the textual requirements that is largely self-explanatory, we found that extensive examples were unnecessary.

Instead, we provided a schema illustrating the expected code structure. This schema included functions for validating each constraint, with a return value of type string.

The model, guided by this schema, automated the remainder of the code generation process with high accuracy (96%). This result highlights the efficacy of our approach in leveraging minimal input to achieve reliable outcomes.

## J    Implementation Details

The CDMS code was implemented using Python 3.10. Chemistry knowledge for data collection and inspection was powered by RDKitLandrum & contributors (2024) version 2022.9.4, a Python toolkit enabling real-time querying of chemical components. However, we observed that the generated inspector code occasionally relied on PubChempyKim et al. (2023), and deepchemdee (2016), but we cannot be sure which package it used in every scenario.

For model deployments, we utilized Azure OpenAI for GPT-4o and Qwen, Anthropic's API for Claude, and NVIDIA NIM for LLaMA.

Inference for the whole dataset costs less than 40$ using the AzureOpenAI and NVIDIA NIM.

## K    Molecule Sampling

In this experiment, we utilized GPT-4o in a zero-shot setting to generate candidate molecules and also as the backbone for inspector-code generation. For each given set of constraints in ChemGen, we first sampled 500 candidate molecules using GPT-4o as the generative model. The generative model was not explicitly instructed to optimize for any specific molecular attribute, such as SAS, QED, or LogP, during this sampling stage. After the candidate molecules were generated, we used GPT-4o to generate the corresponding inspector code, denoted as $M_I$, for the same set of constraints. The generated inspector was then used as a first-stage filter over the 500 sampled molecules.

### K.1    Accuracy

The first-stage filtering pass rate was 46.00%. That is, 46.00% of the 500 zero-shot sampled molecules passed the validation of the generated inspector $M_I$. This number reflects only the acceptance rate of the first filtering phase and should not be interpreted as the end-to-end accuracy of CDMS or as accuracy with respect to the golden-standard inspector code.

### K.2    Molecule Selection

After applying the first-stage filter, we selected molecules from the sampled candidates based on their scores for each metric—SAS, QED, and LogP—using a predictive scoring model. Specifically, we selected the top 10% of the original sampled pool, corresponding to 50 molecules, for each metric. The results indicate that these molecular properties could be effectively optimized through post-generation selection, even though the generative model was not explicitly prompted to optimize them during the initial sampling stage. Furthermore, 44% of the selected molecules passed the generated-inspector filter, suggesting that property-based selection did not substantially reduce the first-stage filtering pass rate.

## L    Molecule synthesis applicable features

To assess the practical applicability of the CDMS framework, we examined its coverage over key aspects of small molecule synthesis and drug discovery. Since CDMS employs code-enabled feedback mechanisms and integrates deep learning models, it is essential to evaluate how well it supports real-world chemical workflows. To this end, we partnered with expert chemists to identify a representative set of features most commonly used in medicinal chemistry. These features span structural, biological, and pharmacological properties. Table 10 summarizes the structural and physicochemical properties, while Table 11 presents attributes related to

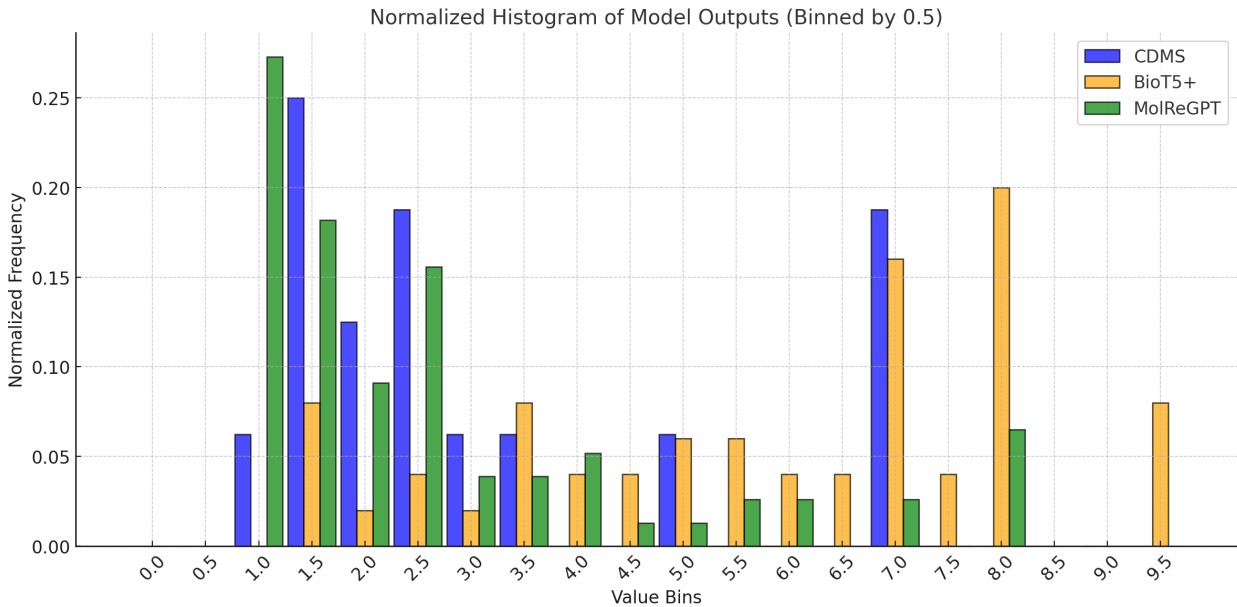

Figure 9: SAS normalized distribution of generated molecules using different generative models

molecular bioactivity. These aspects are further annotated to indicate whether they are computable via code (e.g., using RDKit), predictable using learned models, or require experimental validation.

## M Application of CDMS in Drug Discovery

We evaluated the applicability of CDMS in the domain of drug discovery by instructing the model to optimize specific aspects of the generated molecules, namely SAS, QED, and LogP, while adhering to the constraints provided in each sample from the ChemGen dataset. The inspector model $M_I$ was equipped with a predictive method capable of estimating the score for each of these aspects.

### M.1 Optimization Methodology

The optimization threshold for each aspect was set as the average score achieved by the model when no optimization request was made. This baseline served as a benchmark to assess whether CDMS was capable of improving the specified aspects. The results indicate that the model was indeed successful in optimizing the requested metrics, surpassing the thresholds for SAS, QED, and LogP.

### M.2 Impact on Accuracy

While the optimization improved the requested aspects, a slight decrease in the accuracy of the generated molecules was observed. The accuracy dropped by 3%–5%, depending on the metric, as detailed below:

- **SAS:** The accuracy decreased by 2.9%.

- **QED:** The accuracy decreased by 4.5%.

- **LogP:** The accuracy decreased by 3.2%.

These results suggest that while CDMS effectively optimized the desired aspects, the trade-off in accuracy remained minimal and within acceptable bounds.

| Attribute | How It Can Be Verified |
|---|---|
| **Structural and Physicochemical Properties** | |
| Molecular Formula | Code (100% accurate) |
| Molecular Weight | Code (100% accurate) |
| Exact Mass | Code (100% accurate) |
| Monoisotopic Mass | Code (100% accurate) |
| Formal Charge | Code (100% accurate) |
| Topological Polar Surface Area (TPSA) | Code (100% accurate) |
| Rotatable Bond Count | Code (100% accurate) |
| Hydrogen Bond Donor/Acceptor Count | Code (100% accurate) |
| Complexity Score | Code (100% accurate) |
| Vapor Density (est.) | Code (100% accurate) |
| Oxidation Behavior (functional group level) | Code (100% accurate) |
| Number of Heavy Atoms | Code (100% accurate) |
| Number of Aromatic Rings | Code (100% accurate) |
| Number of Aliphatic Rings | Code (100% accurate) |
| Number of Saturated Rings | Code (100% accurate) |
| Number of Ring Systems | Code (100% accurate) |
| Number of Bridgehead Atoms | Code (100% accurate) |
| Number of Spiro Atoms | Code (100% accurate) |
| Number of Heteroatoms | Code (100% accurate) |
| Number of Nitrogens and Oxygens | Code (100% accurate) |
| Number of Valence Electrons | Code (100% accurate) |
| Number of Radical Electrons | Code (100% accurate) |
| Fraction of Csp3 Carbons | Code (100% accurate) |
| Ring Count | Code (100% accurate) |
| **Predicted or Approximate Properties** | |
| XLogP3 (predicted LogP) | Code (high probability) |
| 1H NMR Spectrum (predicted) | Code (high probability) |
| Physical State (approx.) | Code (high probability) |
| Primary Hazard Estimation (QSAR-based) | Code (high probability) |
| Volatility with Steam | Code (high probability) |
| Boiling Point (predicted) | Code (medium–low accuracy) |
| Melting Point (predicted) | Code (medium–low accuracy) |
| Solubility in Water (predicted) | Code (medium–low accuracy) |
| pKa (predicted) | Code (medium–low accuracy) |
| Flash Point (predicted) | Code (medium–low accuracy) |
| Vapor Pressure (predicted) | Code (medium–low accuracy) |
| **Properties Requiring Experimental Validation** | |
| LogP (experimental) | Lab required |
| Boiling Point (true) | Lab required |
| Melting Point (true) | Lab required |
| Solubility (true) | Lab required |
| pKa (accurate value) | Lab required |
| Flash Point (certified) | Lab required |
| Vapor Pressure (precise) | Lab required |
| Reduction Behavior (e.g., silver nitrate test) | Lab required |
| Shelf Stability | Lab required |
| 1H NMR Spectrum (measured) | Lab required |
| MS Spectrum | Lab required |
| IR Spectrum | Lab required |

Table 10: Verification modes of synthesis-relevant molecular attributes

| Biological/Pharmacological Aspect | How It Can Be Verified |
|---|---|
| Blood–Brain Barrier Permeability | Code (high probability) |
| General Toxicity (e.g., Tox21) | Code (high probability) |
| High-Throughput Toxicity (e.g., ToxCast) | Code (high probability) |
| Virtual Screening Effectiveness (e.g., MUV) | Code (high probability) |
| HIV Inhibition Activity | Code (high probability) |
| Beta-Secretase Inhibition (BACE) | Code (high probability) |
| Side Effect Likelihood (e.g., SIDER) | Code (medium–low accuracy) |
| Clinical Toxicity Risk (e.g., ClinTox) | Code (medium–low accuracy) |
| Aggregate Bioactivity Score | Code (high probability) |

Table 11: Verification modes of biological and pharmacological molecular properties

