# OpenReview forum: "Enhancing Large Language Models for Constraint-Driven Molecular Generation and Beyond"
_TMLR — Under review for TMLR_

### Review · Reviewer_HTEf · 2026-06-15

**Summary Of Contributions:**

## Summary

The authors introduce Code-Driven Molecular Synthesis (CDMS), a model-agnostic framework that embeds code-based feedback in an iterative, large language model (LLM)-based molecule generation process. CDMS does not require any model pretraining. Instead, its code generation process is based on a single-shot example. The authors introduce a novel dataset to evaluate CDMS and compare its performance against several state-of-the art baselines. Not only does CDMS outperform the selected (representative) baselines, but the authors also demonstrate that the molecules generated by CDMS are highly diverse. While the high diversity makes CDMS a good framework for de novo molecular synthesis, the authors also highlight the capabilities of the framework within other domains, such as natural language processing.

---
I thank the authors for their interesting proposal. I first summarize the strengths of the proposal, as well as my questions and concerns. The full evaluation is shared below.

### Strengths
* **Relevance**: Currently, there is a surge of interest in methods that accelerate scientific discovery with machine learning and deep learning. The proposed framework forms an appropriate contribution to the field. The framework and new dataset are expected to be of interest to plenty researchers.
* **Coverage**: The authors introduce both a new framework for de novo molecule generation, as well as a novel dataset for evaluating generative methods.
* **Baselines**: The authors compare their proposed framework against an elaborate and representative set of baselines including zero-shot generative models, iterative refinement models and task-specific trained models.
* **Comparison with SWE-bench**: I really appreciate the comparison with the SWE-bench performance as presented in Figure 5. This helps set expectations for the performance of CDMS when using novel models.

### Weaknesses and questions

**Weaknesses**
* **Support of claims made**: Not all claims made in the introduction are sufficiently supported by empirical evidence or backed by the literature.
* **Evaluation clarity**: While the empirical results as presented in Table 5.1 are clear, the manuscript could benefit from an elaboration on the metrics used in Section 6. The exact methodology for evaluating the synthesizability (SAS) and drug likeness (QED) of the generated molecules. Appendix K.2 briefly introduces “a predictive scoring model” on that matter but lacks sufficient detail.

**Minor comments**
* **Writing consistency**: The terms and components of CDMS are not introduced consistently across the manuscript. For example, the _Inspector Model_ is sometimes italic (Section 3.1), and sometimes not (Section 3, intro). Similarly, CDMS is sometimes written in bold, other times in italic. It is unclear what the asterisk in Table 1 (CDMS -> CoT) refers to. The authors do not clarify the abbreviations QED and SAS  in Section 6.1
* **References**: Some of the references are not properly capitalized (e.g., Bai et al., 2022, Grattafiori et al., 2024, Li et al., 2023, Li et al., 2024). The reference for the October 8 version of Claude (Anthropic, 2023) seems wrongly parsed in the URL field.
* **Formatting**: The code in Appendix A is incorrectly indented in line 1. The code is also interrupted by Table 4.

**Questions**

* **Dataset Construction**: Could the authors provide a breakdown of the number of requirements per attribute category included in the dataset?
* **Molecular diversity**: Could the authors elaborate how the Tanimoto-based molecular diversity is analysed?
* **Improvement of refinement attempts**: The accuracy of generated answers versus the number of refinement attempts is currently presented in Figure 4. However, could the authors also include an analysis of the fraction of false positives and false negatives as identified by the inspector when compared to the golden validator? It would be very interesting to learn whether there may be categories (see the question on **dataset construction) that are harder to validate than others. This would also provide future practitioners with more insight in the strengths and weaknesses of the proposed algorithm.

**Audience:**

Yes

**Audience Explanation:**

Yes. Currently, there is a surge of interest in methods that accelerate scientific discovery with machine learning and deep learning. The proposed framework forms an appropriate contribution to the field. The framework and new dataset are expected to be of interest to plenty researchers.

**Broader Impact Concerns:**

There seem to be no ethical implications of the work that go beyond the implications associated with generative models, such as large language models. Therefore, I consider adding a broader impact statement optional.

**Claims And Evidence:**

Yes

**Claims Explanation:**

Yes. The main claims regarding the performance of the introduced framework are sufficiently supported by experimental evidence.

However, some claims made in the introduction and related work could be better embedded in the existing literature. For example, the authors state that “Recent advancements have introduced end-to-end generative models to streamline molecular candidate generation. However, these models often generate molecules that do not adhere to the chemists’ requirements or are impractical to synthesize due to unavailable precursors or overly complex synthesis routes.”  (Introduction) without (1) providing any example of such a recent advancements and (2) linking them to the proposed lack of adherence.

**Requested Changes:**

* Improve the literature support of the claims made above.
* Elaborate the evaluation procedure of the QED and SAS-related findings
* Resolving all minor comments and answering the questions as stated above, would in my view strengthen the work.

---

### Review · Reviewer_zaMw · 2026-06-27

**Summary Of Contributions:**

The paper proposes CDMS, an iterative framework for constraint-driven molecular generation. Given a natural-language request, an LLM generates executable inspector code that checks whether a candidate SMILES satisfies the requested constraints. The inspector then provides textual feedback to guide later generations. The paper also introduces ChemGen, a benchmark of molecular constraint prompts with expert-written validators.

The main strength is the simple and effective use of executable feedback. The results show large improvements over prompting, Self-Refine/TextGrad-style methods, and several molecule-generation baselines. The ablations are also useful, especially the comparison between binary and textual feedback.

The main weakness is that the paper overstates the chemistry impact. The experiments mainly show satisfaction of computable constraints, not actual synthesis, retrosynthesis, activity, or experimental drug discovery. The reliability of the generated inspector code is also a central concern.

**Audience:**

Yes

**Audience Explanation:**

The paper should interest researchers working on ML for molecular generation. The idea of using persistent executable inspectors for constrained generation is simple and useful.

**Claims And Evidence:**

Yes

**Claims Explanation:**

The claim that programmatic feedback improves constraint satisfaction is well supported by the main results and ablations.

**Requested Changes:**

- Clarify the scope of the claims. The paper should distinguish constraint satisfaction from chemical validity, drug-likeness, and real drug-discovery utility.
- Provide more detail on inspector reliability. Since the generated code drives the method, the paper should report false positives/false negatives against the golden validators and explain how the reported 96% inspector accuracy was measured.
- Add failure cases. Examples where CDMS fails would make the limitations much clearer, especially cases where the generated inspector is wrong or incomplete.
- Clarify whether ChemGen is overly aligned with CDMS. Since the benchmark also relies on executable validators, the paper should explain why this does not unfairly favour the proposed method.
- Figure 1 is low quality and looks really AI-generated. I can easily see several white lines on it. Could you revise it?
- Figure 2 has blue lines that make the text in the box harder to read.

---

### Review · Reviewer_J9a7 · 2026-07-04

**Summary Of Contributions:**

This paper introduces Code-Driven Molecular Synthesis (CDMS), an iterative framework for constraint-driven molecule generation. Given a natural-language specification of molecular requirements, an LLM first generates executable Python/RDKit-based inspector code. This generated inspector is kept fixed throughout the refinement process and is repeatedly applied to candidate SMILES strings produced by a generator model. When a candidate violates one or more constraints, the inspector returns natural-language corrective feedback, referred to by the authors as “Programmatic Feedback Gradients,” which is appended to the subsequent generator prompt.

The paper also introduces ChemGen, a benchmark of 779 natural-language molecular-constraint examples paired with expert-authored golden validation code. The experiments report substantial improvements in constraint satisfaction over zero-shot prompting, chain-of-thought prompting, few-shot prompting, Self-Refine, TextGrad, MolReGPT/MolRe-X, ChemLLM, mCLM, and fine-tuned BioT5+. The paper further argues that the same idea generalizes beyond molecular generation to constrained text generation and structured-data generation, and includes additional experiments on drug-like molecule generation and property optimization.

The main strengths are the following. First, compiling natural-language constraints into executable validators is a practically useful and conceptually clean idea. Second, the one-to-many evaluation setting is appropriate for constraint-driven molecular generation, where there is often no unique target molecule. Third, the iterative feedback loop is intuitive and likely beneficial in practice. Fourth, the paper includes several relevant ablations, including binary versus textual feedback and the use of weaker inspector-generation models. Finally, releasing a constraint-driven benchmark with golden validators could be valuable for future work on tool-augmented scientific generation.

The main weaknesses are also substantial. The strongest empirical claims depend on generated inspector code whose correctness is only partially validated. The baseline comparisons do not clearly equalize tool access, verifier access, sampling budget, or iteration budget. Several application-level claims, especially those concerning drug discovery and state-of-the-art performance, are stronger than the evidence currently supports. The formal presentation is relatively thin, and the term “gradients” is potentially misleading because no mathematical gradient or differentiable update is defined. Finally, the broader-impact and safety discussion is underdeveloped for a method that can guide molecular design.

**Additional Comments:**

- The framework is interesting and potentially useful, but the current evidence supports a narrower conclusion: executable feedback improves satisfaction of computable descriptor/SMARTS constraints on a newly constructed benchmark.
- The prompt states that “There is always a feasible solution,” but real users may provide infeasible or mutually inconsistent constraints. The paper should test or discuss infeasible-constraint detection.
- The limitations section is useful, but the abstract and introduction should also use more careful language about chemical validity and drug-discovery relevance.
- The paper should report whether generated inspectors and golden inspectors use the same RDKit functions and SMARTS patterns. If so, performance may partly reflect alignment to RDKit-coded constraints rather than broader chemical understanding.
- The generalization claim in “and beyond” is plausible but currently supported mainly by synthetic tasks; it should be framed as preliminary evidence.

**Audience:**

Yes

**Audience Explanation:**

The paper is likely to interest TMLR readers working on tool-augmented generation, constrained generation, neuro-symbolic methods, LLM agents, and scientific applications of language models. The idea of compiling natural-language constraints into executable validators and using them as iterative feedback is broadly relevant beyond molecular generation. ChemGen could also become a useful benchmark if the dataset, golden validators, prompts, raw outputs, splits, and evaluation code are released in a reproducible form. That said, the paper should more carefully scope its conclusions and strengthen its evidence before the broader scientific and drug-discovery claims are convincing.

**Broader Impact Concerns:**

The paper needs a stronger broader-impact discussion. Constraint-driven molecular generation is dual-use: the same mechanism that helps satisfy benign medicinal-chemistry constraints could also help satisfy constraints for toxic, hazardous, controlled, or environmentally harmful compounds. The authors should discuss misuse scenarios, safety filters, restricted-release considerations, and whether the benchmark includes or excludes dangerous functional groups or toxicity-related objectives.

There is also a code-execution risk. CDMS executes LLM-generated Python inspector code. If the framework is released as an agentic tool, generated code should be sandboxed, denied network and filesystem access by default, and constrained to safe chemistry libraries with strict resource and timeout limits.

Finally, the paper should avoid overstating practical drug-discovery impact. Optimizing proxy scores such as SAS, QED, and LogP may create misleading confidence, especially for non-expert users. The broader-impact statement should explicitly state that generated molecules require expert review and laboratory validation before any real-world use.

**Claims And Evidence:**

No

**Claims Explanation:**

The overall direction is promising, and the reported results suggest that executable feedback can substantially improve constraint satisfaction. However, several central claims are not yet supported by sufficiently clear, complete, and convincing evidence.

First, the paper should more clearly distinguish among three different objects: the original natural-language constraints, the generated inspector used inside CDMS, and the expert-authored golden validator used for final evaluation. This distinction is central to the method. The paper reports that manual validation found 96% of generated inspector snippets to accurately represent the original prompts, but it does not provide the sample size, sampling procedure, annotation protocol, failure categories, inter-annotator agreement, or a breakdown by constraint type. Since CDMS is guided by generated code, even a small number of systematic inspector errors could steer the generator toward molecules that satisfy the generated inspector while violating the intended chemical requirement. The paper should therefore report generated-inspector accuracy against the golden validators at the level of individual constraints and constraint categories, rather than only reporting end-to-end success rates.

Second, the fairness of the baseline comparisons is not fully established. CDMS receives executable validation feedback at every refinement step, whereas many baselines appear to receive only prompting or self-generated feedback. TextGrad is included, but the adaptation is only briefly described. The comparison would be much more convincing if all iterative baselines were given the same RDKit/golden-validator signals under the same model-call budget, or if the paper explicitly separated “LLM-only” baselines from “tool-feedback” baselines. The `k`-attempts baseline is also under-specified: it is unclear whether it uses the same total number of generator calls as CDMS, how candidates are selected, and whether validator-based selection is allowed.

Third, the state-of-the-art claim should be more carefully scoped. ChemGen is a newly introduced benchmark, and many of the compared molecule-generation models were not designed for this exact one-to-many constraint-satisfaction setting. It is reasonable to claim that CDMS is strong on ChemGen under the proposed evaluation protocol, but a broader state-of-the-art claim is difficult to assess without additional external benchmarks and better-matched baselines. Comparisons to ChemLLM, BioT5+, mCLM, and MolReGPT should also clarify whether these models were prompted or fine-tuned in a way that is well suited to constraint satisfaction rather than single-target molecule reconstruction.

Fourth, the chemistry evidence is narrower than the application framing suggests. Passing RDKit descriptor checks and SMARTS-pattern checks does not imply that a molecule is stable, synthetically feasible, safe, experimentally meaningful, or drug-like. The paper acknowledges molecular stability as a limitation, but the abstract and introduction repeatedly invoke de novo synthesis, drug discovery, and practical chemical design. SAS, QED, and XLogP are useful proxy scores, but they do not by themselves establish synthesizability or downstream biological utility. Claims about producing “real” or practically useful chemical compounds should therefore be weakened unless stronger chemical validation or expert review is added.

Fifth, the drug-like generation experiment in Table 3 is not clearly comparable to the end-to-end baselines. Appendix K indicates that 500 candidate molecules are sampled and that the top 10% are selected after filtering/scoring. This gives CDMS a much larger candidate-generation and selection budget than a single-pass end-to-end generator. The result may still be interesting as a property-optimization demonstration, but it should not be presented as a direct comparison unless all methods are evaluated under the same sampling, filtering, and selection budget. The paper should also report how many selected molecules satisfy the expert-authored golden constraints, not only the generated-inspector filter.

Sixth, statistical support is incomplete. Table 1 states that McNemar's test was used, but the paper does not report exact p-values, confidence intervals, multiple-comparison handling, or which pairwise comparisons were tested. Given the relatively small dataset size and the many model/prompt variants, uncertainty estimates would be important for interpreting differences such as 48.09 versus 51.47 or 52.94 versus 53.92.

Finally, the term “Programmatic Feedback Gradients” is not formally justified. The feedback produced by the inspector is textual, rule-based corrective feedback. No derivative, local linearization, gradient estimator, or optimization geometry is defined. The term may be acceptable as a metaphor, but the paper should avoid language that suggests differentiable or gradient-based optimization unless a gradient-like object is actually formalized.

**Requested Changes:**

1. **Clarify the complete evaluation pipeline.** Precisely define when the generated inspector is used, when the golden validator is used, and how final success is computed. The method section should explicitly track the original constraints, memory, candidate output, generated inspector, and golden validator. Algorithm 1 currently omits the memory component and makes `GenModel(feedback)` appear to receive only the latest feedback rather than the original task plus the interaction history.

2. **Rigorously validate generated inspectors.** Report inspector correctness against the golden validators by constraint type, including descriptor constraints, functional-group constraints, ring constraints, invalid SMILES handling, salts/ions, tautomers, and edge cases. The 96% manual-validation claim needs sample size, annotation rubric, annotator expertise, inter-annotator agreement, and representative failure examples.

3. **Add equal-budget tool-feedback baselines.** Compare CDMS against baselines that receive the same validation tools and model-call budget. Useful baselines would include best-of-`k` with golden-validator selection, best-of-`k` with generated-inspector selection, Self-Refine with RDKit/golden feedback, TextGrad with the same validator and iteration count, and direct repair prompting using the exact failed constraints.

4. **Scope the state-of-the-art claim.** Replace broad state-of-the-art language with a claim specific to ChemGen and the paper's evaluation protocol, unless additional external benchmarks and better-matched comparisons are added.

5. **Report uncertainty and significance transparently.** Provide confidence intervals for all success rates, exact McNemar-test details, multiple-comparison corrections if applicable, and preferably results over repeated LLM sampling runs. If repeated closed-model runs are too expensive, bootstrap confidence intervals over examples would still be useful.

6. **Decompose performance by difficulty and constraint type.** Report success rates as a function of the number of constraints, operator type, descriptor category, SMARTS pattern, and estimated PubChem candidate count. This would clarify whether CDMS solves genuinely hard multi-constraint cases or mostly simpler descriptor cases.

7. **Clarify dataset construction and leakage risks.** Since each ChemGen example is derived from at least one PubChem molecule satisfying the constraints, clarify whether the textual constraints include near-identifying numeric values that make the source molecule easy to recover. If any model is fine-tuned, provide train/validation/test splits and ensure that related molecules or near-duplicate constraint sets do not leak across splits.

8. **Strengthen or weaken chemical-validity claims.** Distinguish among valid SMILES, RDKit-sanitizable molecules, known PubChem compounds, stable molecules, synthetically accessible molecules, and plausible drug-like candidates. If practical synthesis relevance is claimed, stronger expert review or chemical validation is needed. Otherwise, the paper should state that CDMS optimizes computable proxy constraints.

9. **Make the Table 3 comparison fair.** Either give all methods the same 500-candidate sampling plus top-10% selection procedure, or present this experiment as an additional CDMS property-optimization demonstration rather than a direct comparison to end-to-end generators.

10. **Formalize or rename “Programmatic Feedback Gradients.”** If “gradient” is intended literally, define an objective, search space, and gradient-like update. If it is intended metaphorically, a clearer term such as “programmatic corrective feedback” or “executable validator feedback” would be preferable.

11. **Improve reproducibility.** Release ChemGen, golden validators, generated inspectors used in experiments, prompts, model versions, decoding parameters, random seeds, raw outputs, invalid-output rates, runtime/cost, and the exact evaluation scripts. Closed-model API versions should be pinned as precisely as possible.

12. **Tighten the generalization experiments.** TextGen and StructuredGen are useful synthetic tests, but the paper should report dataset sizes, generation procedures, difficulty splits, verifier construction, and quality-control procedures. The human quality evaluation should include sample size, annotator instructions, agreement statistics, and significance tests.